# Unmet Needs of Parents of Children with Urea Cycle Disorders

**DOI:** 10.3390/children9050712

**Published:** 2022-05-12

**Authors:** Mara Scharping, Heiko Brennenstuhl, Sven F. Garbade, Beate Wild, Roland Posset, Matthias Zielonka, Stefan Kölker, Markus W. Haun, Thomas Opladen

**Affiliations:** 1Division of Child Neurology and Metabolic Medicine, Centre for Child and Adolescent Medicine, University Hospital Heidelberg, 69120 Heidelberg, Germany; mara.scharping@med.uni-heidelberg.de (M.S.); heiko.brennenstuhl@med.uni-heidelberg.de (H.B.); sven.garbade@med.uni-heidelberg.de (S.F.G.); roland.posset@med.uni-heidelberg.de (R.P.); matthias.zielonka@med.uni-heidelberg.de (M.Z.); stefan.koelker@med.uni-heidelberg.de (S.K.); 2Department of General Internal Medicine and Psychosomatics, University Hospital Heidelberg, 69120 Heidelberg, Germany; beate.wild@med.uni-heidelberg.de (B.W.); markus.haun@med.uni-heidelberg.de (M.W.H.)

**Keywords:** family burden, parental need, urea cycle disorders, E-IMD, inherited metabolic diseases

## Abstract

(1) Background: Phenotypic diversity and long-term health outcomes of individuals with urea cycle disorders (UCDs) have been described in detail. However, there is limited information on the burden on affected families. (2) Methods: To evaluate the family burden in parents with children suffering from UCDs, we used validated questionnaires. Socio-demographic characteristics were evaluated, and an adapted version of the Parental Need Scale for Rare Diseases questionnaire was used. The survey was conducted in families of UCD patients cared for at the University Children’s Hospital Heidelberg. (3) Results: From April to November 2021, 59 participants were interviewed (mothers *n* = 34, fathers *n* = 25). The affected patients most frequently suffered from ornithine transcarbamylase deficiency (OTC-D) (female *n* = 12, male *n* = 12), followed by argininosuccinate synthetase deficiency (ASS-D, *n* = 13) and argininosuccinate lyase deficiency (ASL-D, *n* = 8). About one-third of the participants were “dissatisfied” or “extremely dissatisfied” with health professionals’ disease knowledge. In addition, 30% of the participants reported a medium or high need for “additional information on the development of their children”, and 44% reported a medium or high need “for information on available services”. A majority of 68% reported a need for additional support regarding services such as support groups (42%) or psychological counseling (29%). (4) Conclusions: Our study indicates that there is an unmet need for sufficient information about the development of children with UCDs, as well as for information about available support services for families with UCD patients. Furthermore, the results highlight the importance of establishing or improving family-centered care approaches. This pilot study may serve as a template for the assessment of the family burden associated with other inherited metabolic diseases.

## 1. Introduction

Inherited metabolic diseases (IMDs) comprise more than 1600 disorders, which are classified into 130 groups based on defects of the respective biochemical metabolic pathways [1]. Most IMDs are rare diseases with low incidences. In the European Union, a disease with an incidence of <1:2000 is considered rare. Patients with rare diseases such as urea cycle disorders (UCD) are cared for in a variety of clinics, most of which are local. The medical expertise and experience of the specific center can vary widely [2]. There can be significant differences in infrastructure, diagnostic procedures, time to diagnosis, strategies and outcomes, and these differences can have a negative impact on health outcomes. In the European Union and other industrialized countries, the care of patients with rare diseases has received special attention in recent years [3]. The newly established European Reference Networks (ERNs) for rare diseases provide a platform for the harmonization of patient care and knowledge exchange across different European countries. Urea cycle disorders comprise a group of IMDs of ureagenesis, which is required for the irreversible elimination of excess nitrogen through the formation of dialyzable urea from ammonium and bicarbonate in periportal hepatocytes. The overall prevalence of UCDs ranges between 1 in 35,000 and 1 in 52,000 in Germany, Austria, and Switzerland, with OTC deficiency (OTC-D) being the most frequent subtype (>50%) [4,5]. Other UCD subtypes are argininosuccinate synthetase deficiency (ASS-D) as the second most frequent disease subtype (19%), followed by argininosuccinate lyase deficiency (ASL-D, 11.5%) and carbamoylphosphate synthetase 1 deficiency (CPS1-D, 4.5%) [5,6]. UCDs are caused by pathogenic variants in genes encoding enzymes or transporters of the urea cycle [7]. The majority of individuals with an UCD manifests with recurrent hyperammonemic episodes precipitated by catabolism, protein-rich meals, medications such as valproic acid, and other trigger factors [8]. The first symptoms may occur during the newborn period or later in life, reflecting the high phenotypic diversity of UCDs [6,9,10]. The degree of enzymatic dysfunction determines the metabolic disease course and its phenotypic severity, i.e., individuals with lower residual enzymatic activity are confronted with higher peak plasma ammonium concentrations at initial presentation and develop more often hyperammonemic decompensations during their disease course [11,12]. Importantly, the level of plasma ammonium at disease manifestation is associated with the neurocognitive outcome, which is most pronounced for mitochondrial UCDs [4,10,13]. Hyperglutaminergic hyperammonemia is the biochemical hallmark of most UCDs. Without immediate therapy, it induces a cascade of synergistically acting mechanisms, such as excitotoxicity, bioenergetic impairment and astrocytic swelling that often results in life-threatening encephalopathy, brain edema, irreversible brain damage and can cause a severe neurodevelopment disorder [14,15]. Hyperammonemia-associated symptoms range from somnolence, nausea, vomiting, liver failure, to seizures, multiorgan failure, acute encephalopathy, and death [16]. The prognosis of UCDs is strongly influenced by the duration of coma and peak ammonia levels in the setting of the initial decompensation [11,12]. In addition, metabolic decompensation can contribute to a worsening neurologic outcome and is therefore a particular burden for parents and caregivers [13,16,17].International recommendations, currently in their second version, have been published for the treatment of UCDs [18]. The long-term management of patients with UCD consists of a low-protein diet, which must be balanced and supplemented to avoid deficiencies of essential amino acids, trace elements or vitamins, and the use of nitrogen scavengers [18]. The acute treatment includes detoxification of ammonia, which often requires extracorporeal hemodialysis, and the use of intravenous drugs that act as nitrogen scavengers [18,19,20,21]. Liver transplantation may be another option [18]. The majority of UCDs are inherited in an autosomal recessive manner [22]. OTC-D is inherited in an X-linked manner [23,24], which leads to severe courses in hemizygous male individuals, while the clinical course in females is extremely variable [23]. So far, UCDs are not part of the German newborn screening program [18]. Currently, the first pilot studies are ongoing to evaluate newborn screening for ASS and ASL deficiency [15,25].

The European registry and network for Intoxication-Type Metabolic Diseases (“E-IMD”) (https://www.eimd-registry.com (accessed on 23 March 2022) gathers comprehensive information on the clinical and biochemical natural history and outcome of UCDs. The goals of the E-IMD include the achievement of a better understanding of the natural history, diagnosis, and treatment options of the diseases, as well as the establishment of guidelines to reduce inequalities in care [2]. However, there is limited information on the psychosocial burden of affected families and their need for support.

Previous studies focused on patients and their challenges associated with a UCD. Specifically, it has been shown that a child’s disease has often an impact on the entire family, as it affects the dynamics of the family and can result in a significant parental burden [26,27,28,29,30]. The time-consuming care for a patient with a UCD requires adjustments for parents, such as a strict dietary control [18], frequent appointments with specialists, and a constant risk of metabolic decompensation. It is assumed that the genetic origin as well as the associated uncertain physical, cognitive, and psychological functioning of the child puts an additional burden on parents [31]. Furthermore, the parental burden can in turn have a negative impact on the affected children and siblings [29]. The difficulties experienced by the parents of children with rare diseases can involve emotional aspects, the relationship with the partner, the own behavior, as well as the process of diagnostics and the challenges of the health care system [32,33]. By understanding the support needs of families with affected children, needs-adapted concepts can be established and further improved, harboring the chance to reduce the family burden in the future [34]. The aim of this work was to evaluate the family burden in parents of children with UCDs in Germany and to assess the parental needs for specific support. Furthermore, we investigated the impact of gender, age, UCD subtype, and income on family burden.

## 2. Materials and Methods

### 2.1. Study Population

The study was conducted as a prospective single-center pilot study at the University Children’s Hospital Heidelberg and as an amendment to the E-IMD study protocol, approved by the Ethics Committee of the University Heidelberg (S-525/2010, approval date is 31 January 2011). All E-IMD UCD patients were identified, duplications caused by siblings or mothers with diagnosed OTC-D were excluded.

In total, 54 E-IMD families including 108 parents were eligible to participate. In addition, one family including 2 parents of a patient with lysinuric protein intolerance (LPI) treated at the University Children’s Hospital Heidelberg and not registered with E-IMD declared interest to participate. Signed consent of all parents to participate in the study was obtained. From April to November 2021, parents were contacted and interviewed by phone or during outpatient visits. All parents were interviewed individually and independently of each other. Out of the 110 eligible parents, 59 (from 46 families) answered the questionnaire (response rate = 53.6%); 22 of the 110 parents screened could not be reached during the data collection period. Other reasons for excluding screened participants were lack of consent (*n* = 15), lack of German language skills (*n* = 10) or death (*n* = 4).

### 2.2. Questionnaire

The study questionnaire used was derived from validated questionnaires: the Parental Needs Survey (PNS) [35] and items for the survey of sociodemographic characteristics “About you being a parent of a child with a rare disease”. In the first part, we collected the sociodemographic data of all participants. For the second part, an adapted version of the PNS was translated into German and used for the interviews. The selection of items from the 108-item PNS questionnaire was conducted by two independent experts with experience in family psychology and therapeutic care of families (MWH/BW). Subsequently, a shortened 28-item version of the PNS was consented. The shortened questionnaire was translated into German by two independent translators (MWH/BW). After backward translation into English by a native Speaker, a German version was compiled. The final version (Appendix A) included 28 items, divided into 6 sections: (1) Understanding the disease (4 items), (2) Working with health professionals (4 items), (3) Financial needs (3 items), (4) Information needs and social, physical, spiritual, and psychological needs (15 items), (5) Need for further support, as well as (6) a free text answer option. The items of the first four categories were answered using a 5-point Likert scale where 1 represented no need for support/full satisfaction, while 5 represented a high need for support/complete dissatisfaction. The need for additional support services was assessed by a list of options to choose from and a free-text field. The questionnaire was implemented using the online survey tool LimeSurvey to ensure correct data entry during the interview and facilitate data extraction.

### 2.3. Statistical Analysis

All survey responses were recorded using LimeSurvey. The complete data were exported as a single csv file. Statistical analyses were performed using R (Version 4.1.0.). Due to the exploratory character of this pilot study, the items were grouped by several categorical variables to reveal potential associations or group-wise differences. For the analysis of gender (mother/father), income (Questionnaire—Item 8: “How do you manage on your available income from all sources?”) and disease subtype (female OTC-D, male OTC-D, ASS-D, ASL-D, CPS1-D, hyperornithinemia–hyperammonemia–homocitrullinuria syndrome (HHH), LPI) specific burden, the results were stratified by subgroups and compared with a X^2^ test. The subgroups female OTC-D and male OTC-D were additionally compared separately due to the X-linked inheritance of this disorder. For the analysis of age-specific differences, we divided the study population into four groups (0–6 y, 6–12 y, 12–18 y, >18 y) according to the age of the patients. The results were then stratified according to these age groups, and a X^2^ test was applied. No a priory hypotheses were tested; therefore, the *p*-values should be regarded as descriptive values. Due to the explanatory approach of our analyses, we omitted controlling the family-wise error rate.

## 3. Results

### 3.1. Sociodemographic Data

Fifty-nine parents (mothers *n* = 34, fathers *n* = 25) of 50 patients with UCDs participated in this study. OTC-D was the most frequent disease subtype, with *n* = 12 male and *n* = 12 female patients, followed by argininosuccinate synthetase deficiency (ASS-D, *n* = 13) and argininosuccinate lyase deficiency (ASL-D, *n* = 8). The frequency of the subtypes was representative of the affected population [5]. The mean age of the parents was 46.1 years (SD = 12.0, range 24–71 years), the mean age of the patients was 15.7 years (SD = 9.9, range 0–47 years). Most parents reported being married (*n* = 44). The most frequently reported highest educational qualification was lower secondary education (“Hauptschulabschluss”) (*n* = 18), followed by high school (“Realschulabschluss”) (*n* = 16) and high school graduate (“Abitur”) (*n* = 13). Most participants reported being in permanent employment (*n* = 38) or homekeeper/retired (*n* = 17). In addition, 73% of the participants stated that they could “easily” manage on their available income, 24% reported “not bad”, and 3% reported “difficulties some of the time”. On median, 2–3 people lived in one household (*n* = 54) with ≤2 children (*n* = 52). Most of the participants described their own health as good (*n* = 38) or fair (*n* = 11). About a quarter of the participants reported being consanguineous (*n* = 16 out of 14 families), with 43% being first-grade cousins, and 57% being second-grade cousins.

### 3.2. Parental Needs Survey

Most parents reported feeling very confident or confident about understanding the disease and explaining it to others (Part 1, Items 1–4). The majority of parents were extremely satisfied or satisfied with working with health professionals and the overall support (Part 2, Items 1–3). About half of the parents reported to feel extremely satisfied (37%, *n* = 22) or satisfied (12%, *n* = 7) with the health professionals’ knowledge about their child’s disease, while about one-fourth reported dissatisfaction (14%, *n* = 8) or extreme dissatisfaction (14%, *n* = 8). This included health professionals from the university hospital as well as from non-specialized institutions.

Most participants indicated that they could easily afford paying for medical care/therapy (75%, *n* = 44) or paying for babysitting/short-term care (76%, *n* = 45). Approximately half of the parents (53%, *n* = 31) reported that they could easily pay for special equipment or clothing, while only few of the participants stated that they had difficulties (14%, *n* = 8) or could not afford special equipment or clothing (12%, *n* = 7). Despite expressed confidence about understanding the disease, about one-third reported a medium (22%, *n* = 13) or high (8%, *n* = 5) need for additional information on the growth and development of their child. Furthermore, the participants reported a medium (22%, *n* = 13) or high (22%, *n* = 13) need for information on current or future services available for their child.

The parents also shared a need for additional supporting services. In fact, 14% (*n* = 8) of them indicated a medium, and 19% (*n* = 11) a high need for support in finding a suitable caretaker; 69% (*n* = 41) of the participants requested no additional need for support regarding the reconciliation of work and family life, while 22% (*n* = 13) reported a high need. About one-third indicated a need (medium = 15%, *n* = 9; high need = 17%, *n* = 10) to talk to other affected families. Most parents reported satisfaction or no additional need for support regarding the relationship with their partners or the siblings of the affected child (Part 4, Item 8 + 9). Only a few participants stated a high need of support for their insomnia (12%, *n* = 7), fatigue (5%, *n* = 3), loss of appetite (2%, *n* = 1) or in finding meaning in the situation (3%, *n* = 2). In contrast, about one-third indicated a medium (15%, *n* = 9) or high (14%, *n* = 8) need of support for their feeling of physical exhaustion.

Moreover, the participants shared a need for support (medium = 15%, *n* = 9; high = 15%, *n* = 9) with feeling useless, powerless, and helpless. Most parents explained no need of support for their communication with health professionals (does not apply = 14%, *n* = 8; satisfied = 59%, *n* = 35). When asked whether they had needs for any further services, 42% (*n* = 25) of the participants reported a need for additional support groups, and 29% (*n* = 17) indicated a need for additional psychological support. Less required additional services were financial counseling (22%, *n* = 13), marriage counseling (20%, *n* = 12), genetic counseling (19%, *n* = 11) and social work (12%, *n* = 7). Overall, 68% (*n* = 40) of the participants reported an unmet need regarding one or more of the listed services.

### 3.3. Gender and Income

Stratification of the data set by participant gender showed a difference (X^2^(4) = 10.9; *p* ≤ 0.05) in terms of satisfaction with “having a consistent team of health professionals who take responsibility for the overall health of my child”: overall, mothers reported to be more satisfied than fathers.

Item 8 of the sociodemographic data collection asks participants to self-assess their ability to manage their situation with the financial income available to them. Looking at the data after stratification, differences were found in three sections: “Financial needs” revealed differences in the need of support with paying for medical care/therapy (X^2^(8) = 16.4; *p* ≤ 0.05) and with paying for special equipment/clothing (X^2^(8) = 15.7; *p* ≤ 0.05). Both items revealed that participants who indicated to have difficulties managing their available household income more often reported a high need for support. The section “Information needs and social, physical, spiritual and psychological needs” showed differences in the need for support regarding the physical and psychological needs. For physical needs such as “feeling of physical exhaustion” (X^2^(8) = 19.0; *p* ≤ 0.01), “insomnia” (X^2^(8) = 25.3; *p* ≤ 0.001) and “loss of appetite” (X^2^(8) = 35.9; *p* ≤ 0.001) and psychological needs like “the need to speak to other parents” (X^2^(8) = 16.5; *p* ≤ 0.05), “finding meaning in the situation” (X^2^(8) = 19.9; *p* ≤ 0.01) and “feeling useless, powerless and helpless” (X^2^(8) = 18.3; *p* ≤ 0.05), our data showed that parents with difficulties managing their available household income more often reported high needs for support. Further, we studied whether there was a difference in the need for additional supporting services. Respondents with higher financial needs more often reported a need for marriage counseling (X^2^(2) = 7.5; *p* ≤ 0.05), financial counseling (X^2^(2) = 15.0; *p* ≤ 0.001), and support groups (X^2^(2) = 10.2; *p* ≤ 0.01); the reported overall need of support for the listed services was also higher (X^2^(2) = 6.8; *p* ≤ 0.05).

### 3.4. Disease Subtypes

The results revealed some differences in all sections, for the data stratified by disease subtype. In the first section, the results showed a difference in the understanding of the disease in items 1 “Teach my child about the disease” (X^2^(18) = 29.5; *p* ≤ 0.05) and 4 “Explain my child’s disease to my parents or relatives” (X^2^(24) = 51.5; *p* ≤ 0.001). Parents of patients with CPS1-D, HHH syndrome or LPI stated more often that they did not feel confident. Parents of patients with HHH syndrome more often reported dissatisfaction regarding “Feeling that you are part of a health care team looking after your child“ (X^2^(24) = 37.7; *p* ≤ 0.05). Furthermore, parents of patients with HHH syndrome or LPI more often reported a high need for financial support with “Paying for special equipment or special clothes” (X^2^(24) = 36.6; *p*≤ 0.05). In the fourth section, the data showed a difference in support needs for item 8, “Relationship with my child’s siblings” (X^2^(18) = 29.2; *p* ≤ 0.05), with parents of patients with ASL-D more often reporting high support needs. There was a difference in the need for support for items 13, “Finding meaning in the situation” (X^2^(24) = 66.0; *p* ≤ 0.001), and 15, “Feeling useless, powerless and helpless” (X^2^(24) = 42.2; *p* ≤ 0.01). Parents of patients with CPS1-D, HHH syndrome or LPI reported a high need more often. The results also showed a difference in the need for additional support services, especially for marriage counseling (X^2^(6) = 13.4; *p* ≤ 0.05), psychological counseling (X^2^(6) = 13.1; *p* ≤ 0.05), support groups (X^2^(6) = 15.9; *p* ≤ 0.01), genetic counseling (X^2^(6) = 15.6; *p* ≤ 0.05) and social work (X^2^(6) = 16.8; *p* ≤ 0.01). The comparison within the OTC group between parents of male OTC-D and female OTC-D patients showed a difference in the need for additional genetic counseling (X^2^(1) = 4.5; *p* ≤ 0.05). Parents of female OTC-D patients indicated a need for genetic counseling more often than parents of male OTC-D patients.

### 3.5. Age of the Patients

For age stratification, the study population was divided into four age groups, each spanning 6 years, based on the age of the patients. The groups of parents interviewed were distributed as follows: 1 (0–6 years): 10 individuals; 2 (6–12 years): 19 individuals; 3 (12–18 years): 12 individuals; 4 (>18 years): 18 individuals. Group 1 represented parents of young patients, group 2 parents of primary school patients, group 3 parents of adolescents, and group 4 parents of adult patients. The results showed, when stratified by age group, differences in the need of support. Parents of younger patients (Group 1 + 3) reported a higher need regarding “Speaking to health professionals” (X^2^(12) = 23.3; *p* ≤ 0.05). In addition, the data revealed a difference in the need for additional services such as marriage counseling (X^2^(3) = 10.1; *p* ≤.05), psychological counseling (X^2^(3) = 17.6; *p* ≤ 0.001) and genetic counseling (X^2^(3) = 12.0; *p* ≤ 0.01). Also in this case, the parents of patients of a younger age (group 1–3) reported more often a need for support. Furthermore, the parents of older patients (group 4) more often indicated no further need for support (X^2^(3) = 10.5; *p* ≤ 0.05).

## 4. Discussion

The aim of this observational, single-center pilot study was the assessment of family burden and need for support in parents of patients with UCDs. The study revealed a consistent need for additional support and information, mainly in the fields of working with healthcare professionals, information, supporting services, and an income-/disease subtype-specific burden.

Our data did show that needs pertaining to understanding the disease were actually met. Previous studies on the support needs of parents with chronically ill children showed a gender difference and a higher burden among women [33], but the results of the present study did not show any difference in this respect (Section 3.3). In addition, the majority of respondents to surveys asking about burden and coping with sick children are usually mothers, as they often share the greater share of parenting responsibilities [29,36]. In our study, about half of the respondents were male (Table 1).

### 4.1. Working with Health Professionals

The complexity of IMDs often requires treatment by specialized multidisciplinary teams [37,38]. It has been shown before that medical care for IMDs outside of specialized centers can be unsatisfactory, especially in emergency situations [37]. On the one hand, our results revealed that most parents were extremely satisfied or satisfied with working with health professionals and the overall support (Figure 1b). In this context, it is important to consider that the participants in this single-center pilot study were cared for by the Department of Pediatric Neurology and Metabolic Medicine at the Centre for Child and Adolescent Medicine University of Heidelberg, which employs IMD specialists. On the other hand, about one-third indicated a dissatisfaction with the health professionals’ knowledge about their child’s disease (Figure 1b). This dissatisfaction might be related to the lack of information on the disease and therapeutic options, as difficulties caused by a lack of knowledge about the disease and a lack of disease-specific support services are common for rare diseases [34]. At the same time, low awareness on rare diseases is often associated with delays in diagnosis and treatment, which in turn leads to an increased burden [33]. In particular, non-specialized health professionals often lack the necessary experience and knowledge [39]. Therefore, there is a need to raise awareness of rare metabolic disorders such as UCDs and improve the medical care outside of specialized facilities. For this, low-threshold collaborations within specialized networks on a national or even international level can be helpful. Patient representatives should be included in these networks. Further studies following this pilot study should further analyze the conclusions mentioned above and the differences between different health care institutions through specific surveys.

### 4.2. Disease-Specific Information Needs

Although most participants reported to feel very confident or confident about the understanding of the disease, about one-third reported a medium or high need for additional information on the growth and development of their child, and nearly half reported a medium or high need for information on current or future services available for their child (Figure 1a). This need might be related to the difficulty of arising questions about the disease that cannot be clearly answered, mostly because of a lack of evidence-based information [40]. The need for information, expressed by the parents, underscores the importance of regular appointments and assessments at specialized facilities for the whole family to keep the families informed of the latest developments in their IMD, answer their questions, and respond to emerging needs [36]. At the same time, parents’ needs highlight the importance of the scientific support of medical care through the establishment of natural history studies and deep phenotyping approaches. For rare diseases, patient registries are thought to be key instruments to achieve a sufficient sample size for the evaluation of the clinical course as well as of diagnostic and therapeutic interventions. In addition, the information supply could be improved through the development of quality-controlled web-based databases and information systems accessible through websites or apps, which could facilitate the accessibility to easy-to-understand knowledge for affected families.

### 4.3. Supporting Services

The need for psychological support among parents of children with IMDs is high, while few reported sufficient availability [36]. It has been shown before that, regardless of the type of condition, the parents of chronically ill children are more likely to have limited health-related Quality of Life (QoL) compared to the parents of healthy children [30,31]. In this study, 68% of the participants indicated an unmet need for one or more of the proposed services, especially the need for support groups and psychological counseling. These findings underscore the importance of support services such as parent advocacy groups and strongly suggest their promotion. Due to the rarity of the diseases and their geographical distribution, the parents of affected children often lack contact with a peer group with similar experiences [34]. In recent years, however, the possibilities for networking via the internet have become more popular and versatile and offer opportunities for affected families to connect. This contact can be beneficial through emotional support and additional information, which could support the parents in dealing with the disease [33,39].

### 4.4. Income-/Disease Subtype-Specific Burden

The results showed differences in the support needs when stratified by the disease subtype of the affected patients or the subjective satisfaction with the participants’ income (Section 3.3). A weakness of our study is the small sample size and, particularly, the uneven distribution within the subgroups. Nevertheless, trends can be derived regarding the need for support. Overall, the reported need for financial support was low (Figure 1c). This is presumably due to the fact that in Germany, medical expenses are in most cases completely covered by the national health care system [36]. In contrast, about a quarter of the respondents indicated an additional need for financial counseling. Furthermore, it could be shown that parents who indicated difficulties with managing their income more often indicated a high need for support in the following sections of the questionnaire (Section 3.3). This is in line with previous studies, which showed a financial burden on the parents of children with an IMD, especially those with a required dietary treatment [26,28]. The analysis of the data according to the diagnosis subtype also showed differences (Section 3.4), whereby the unequal and sometimes very small group size of the subgroups must also be considered here. However, a tendency can be shown that the parents of patients with HHH syndrome, CPS1-D or LPI, disorders that have a very low incidence in common (approx. < 1:2,000,000 [5]), more often indicated a high need for support. This may be related to the lack of evidence-based information and peer support [34,40], which may be even more pronounced for these very rare diseases. Stratification by sex of the patients revealed a difference in burden among the parents of patients with OTC-D, as the parents of female patients with OTC-D more frequently reported a need for genetic counseling (Section 3.4). Apart from this, no specific burden was found among the parents of male patients with OTC-D compared with those of female patients with OTC-D, which is surprising, considering the higher severity of the disease course in male patients.

### 4.5. Age-Specific Burden

Due to advances in the diagnosis and therapy options of IMDs, more and more affected children are reaching adulthood, which presents new challenges [27,41,42]. UCD patients usually require lifelong therapy, which is why appropriate care is also necessary in adulthood [41,42,43]. Such transitions are challenging for the health care system, as well as for patients and caregivers [38], and the access to services that promote the transition to an independent adult life of patients with IMDs is limited [36]. This suggests that depending on the age of the patient, varying difficulties and burdens for the parents could be in focus. When stratified by the age groups of the patients, the results showed some differences in the need for additional support services (Section 3.5). Overall, the evaluation showed that parents of older patients in group 3 (12–18 years) and 4 (>18 years) more often had no need for additional support services. Those results are in line with previous studies on the psychological adjustment of parents of chronically ill children, which suggest that good adjustment is possible, but still the risk of poor adjustment is significantly higher than in the general population [29,30,44].

### 4.6. Study Limitations and Strengths

This study is subject to certain limitations. First, the study was planned and conducted as a pilot study and includes patients treated only in one university center in Germany. This consideration neglects possible differences between care in different sectors of the health care system. Furthermore, national differences in care and support, e.g., due to different health care systems, must be assumed, so follow-up studies should include an international perspective. The European collaboration of the E-IMD network may be helpful for this. In addition, because of the rarity of UCD-associated diseases, only single families could be studied for some enzyme defects. Therefore, confirmation of the results in a larger collective would be desirable. Another limitation is the partly deficient language skills and e-health competence, which may have influenced the accuracy of the responses of the respective participants. Furthermore, it must be considered that the study was conducted during the COVID-19 pandemic. The effects of the pandemic and changes in the daily lives as well as in the healthcare systems may have put an additional burden on the families during the study period [45] and biased the results of this study.

A strength of this study is found in the fact that despite being single-center study on a rare disease, a high number of participants, especially of male responders, was achieved. The study population covered a wide age range of UCD subtypes. Despite an overall small study group, trends could be identified. Through the multidisciplinary team of contributors, versatile views and interpretations of the results could be achieved. The use and adaptation of validated questionnaires and the online implementation makes the survey easily applicable to other disorders in multi-center studies.

## 5. Conclusions

In conclusion, the results of this study demonstrate the importance of developing and piloting family-centered care approaches. The burden situation and support requirements of families with affected patients still do not seem to be fully assessed by medical professionals and should therefore be regularly and thoroughly reviewed. Knowledge of the disease process and the latest information about it should be regularly shared with families. This pilot study can serve as a template for assessing the family burden of inherited metabolic diseases. In a future study, a multicenter approach could be taken to increase the number of participants to also identify site-specific gaps in care to capture regional differences. A structured international comparison of different health care systems, e.g., in the context of the European reference network MetabERN, could also be considered.

## Figures and Tables

**Figure 1 children-09-00712-f001:**
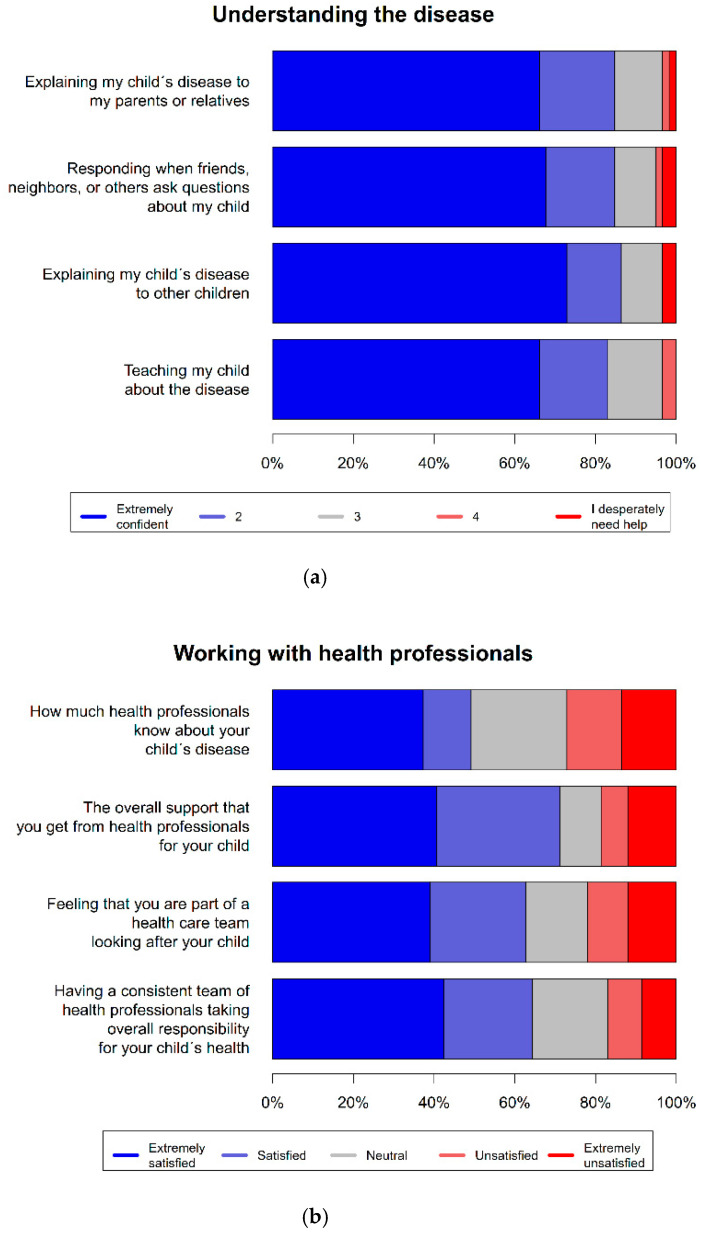
(**a**) Part 1: Understanding of the disease, (**b**) Part 2: Working with health professionals, (**c**) Part 3: Financial needs, (**d**) Part 4: Informational, social, physical, spiritual and psychological needs. (The answer option “does not apply” in Section 4 is to be understood as “no need” (neither satisfaction with existing services nor need for additional support).

**Table 1 children-09-00712-t001:** Sociodemographic Characteristics.

Participants	*n*	%
Mothers	34	57.6
Fathers	25	42.4
Total	59	100.0
Patients	50	
UCD Subtypes		
ASL-D	8	16.0
ASS-D	13	26.0
CPS1-D	2	4.0
HHH Syndrome	2	4.0
LPI	1	2.0
OTC-D	24	48.0
*Female* OTC-D	12	24.0
*Male* OTC-D	12	24.0
Marital Status		
Divorced	4	6.8
Separated	4	6.8
Member of an unmarried couple	4	6.8
Married	44	74.6
Widowed	3	5.0
Highest Educational Status		
Never attended school or kindergarten	1	1.7
Lower secondary education	18	30.5
High school	16	27.1
High school graduate	13	22.0
College (≤3 Years)	2	3.4
College (>3 Years)	7	11.9
I do not know/Not sure	2	3.4
Employment Status		
Employed for wages	38	64.4
Homekeeper/Retired	17	28.8
Self-employed	4	6.8
How do you manage on your available household income from all sources?	*n*	%
Easily	43	72.9
Not Bad	14	23.7
Difficult some of the times	2	3.4
How many people live in your household?		
<18		
0	16	27.1
1	16	27.1
2	20	33.9
3	5	8.5
4	0	0.0
5	2	3.4
>18		
1	1	1.7
2	41	69.5
3	13	22.0
4	4	6.8
Would you say that in general your health is…		
Excellent	3	5.1
Very Good	4	6.8
Good	38	64.4
Fair	11	18.6
Bad	3	5.1
How many of your children are affected by a rare disease?		
1	51	86.4
2	7	6.8
3	1	1.7
How many biological siblings does your affected child have?		
0	19	32.2
1	25	42.4
2	12	20.3
3	1	1.7
4	2	3.4
Are you and the other parent of the affected child related? If yes, please indicate the degree.		
Yes	16	27.1
First	6	10.2
Second	8	13.6
>Second	2	3.4
No	42	71.2
Do not know	1	1.7

List of abbreviations: OTC-D (ornithine transcarbamylase deficiency) with subgroups *female/male* OTC-D, ASS-D (argininosuccinate synthetase deficiency), ASL-D (argininosuccinate lyase deficiency), CPS1-D (carbamoylphosphate synthetase 1 deficiency), HHH (hyperornithinemia–hyperammonemia–homocitrullinuria syndrome), LPI (lysinuric protein intolerance).

## Data Availability

All data supporting the findings described in this manuscript are not publicly available due to existing data protection laws but are available from the corresponding author (T.O.) upon reasonable request and within the limitations of the informed consent.

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
