# Peer review of "Unmet Needs of Parents of Children with Urea Cycle Disorders"

_children, 2022, doi:10.3390/children9050712_

Round 1

Reviewer 1 Report

This article provides an interesting snapshot of the burden perceived by parents of children and adults diagnosed with rare complex medical conditions, as UCD, followed by a single Centre, of expertise for these RD. 

Below some comments with the intent of increasing the clarity of the manuscript for potential readers.  

Introduction

Authors report the number of IMD as derived from Ferreira et al., 2019. It would be appropriate to quantify the number of UCD according to the same proposed nosology. So that readers can easily understand the proportion of UCD investigated in the present article with respect to the whole group. 

The introduction could benefit from the following actions:

  • Please add other references quantifying the prevalence of UCD at population level
  • please provide some information on the German healthcare system organization and provision of services for patients with rare diseases (RDs)
  • mention how is the current situation regarding newborn screening for these conditions in Germany
  • Please add some information on available treatments for UCD
  • Please add references dealing with the long-term outcomes of UCD which could affect the burden on families
  • Please add reference to studies investigating the impact of the pandemic on the care experience of RDs patients and in patients diagnosed with IMD in particular see Lampe C et al. The impact of COVID-19 on rare metabolic patients and healthcare providers: results from two MetabERN surveys. Orphanet J Rare Dis. 2020 Dec 3;15(1):341.

Section 2.1

  • In the sample described, Authors reported a high percentage of consanguineous parents (nearly one out of 3). I think this datum deserves further specifications, reporting how many of these consanguineous individuals are parents of the same patient.
  • Furthermore, it would be useful to add among the socio-demographic data the country of origin of the parents involved in the study, if possible. It would be interesting for readers to know whether the country of origin of parents might have affected their ability to participate in the study, thus creating a selection bias as some questions deal with language skills, as the capacity of interacting with health professionals or understanding disease issues. 
    Therefore, among the limitations of the study, I think the issue of literacy and e-health literacy 8as the questionnaire had to be filled online) of interviewed parents should be better tackled.  

Section 3 Results

A general comment is that a descriptive analysis should be provided for all the results. Furthermore, it has to be noted that in many cases results consider as statistical unit the parent, but when the stratification is used the groups are in many cases based on variables which refer to the patient, introducing a bias. For example the income, except in the case of single parents is related to the family and depends on its composition. Thus, when considering the financial issue this should be investigated after adjusting for the family composition i.e. number of individuals and /or working situation of the parents.

Figure 4. Many questions have a high % of answers falling in the category “does not apply”. This could be appropriate in some cases (i.e. for single parents “relationship with my partner”), but it is of challenging interpretation for other questions more related to relational needs i.e. need to speak to other parents with similar experiences.

Apart from these limitations, as the findings related to the X2 test  are not reported in the discussion I would consider to better present the descriptive results and/or modify the discussion accordingly.

Minor comment: when describing the study population I think the term patients is more appropriate than children, as there are also some adult patients.

Section 4.5.

The issue of care transition is a particularly topical theme when dealing with IMD.  Please add some references on the care transition issue in IMD and in RD patients  

(i.e. Stepien KM et al. Challenges in Transition From Childhood to Adulthood Care in Rare Metabolic Diseases: Results From the First Multi-Center European Survey. Front Med 2021; 8:652358 

and Mazzucato M et al. The Epidemiology of Transition into Adulthood of Rare Diseases Patients: Results from a Population-Based Registry. Int J Environ Res Public Health. 2018;15(10):2212)

4.6 Study limitations

The issue of parents who refused to participate should be tackled to highlight potential bias in the sample described. The issue of literacy and e-health literacy has been already pointed out in my comments, starting from the n=10 parents reported to lack language skills. There is one additional parent reported as “never attended school”, but who filled the questionnaire.

Another limit to be considered is that results, especially for questions regarding care provision, could depend on the general benefits and pathways available in Germany for these patients.  

Another limit is that answers were collected during the period (April-Niovember 2021). Thus they could have been influenced by the reorganization of care consequent to the pandemic emergency, and by the peculiar feelings emerging in parents of children and adults affected by complex medical conditions as UCD.

All these points should be introduced in the background of the article and then discussed at the end of it, considering the results.

Also the strengths of the study should be mentioned.

Conclusions

The MetabERN is cited. I think this is an interesting perspective for a possible future multicenter approach of similar studies. Nevertheless, as maybe not all readers are familiar with European Reference Network it should be explained what ERNs are, mentioning the attention deserved to collaboration with patients’ associations and their role within them.

Reviewer 2 Report

This paper analyzed the burden of UCD on families and how their needs are covered at different levels, such as information, support services, relationships with the health professionals, and financial needs. These kinds of studies are few in the field of rare diseases and this could be used as a basis for improvements in policies for these patients. The study was performed in one center, based on responses from 59 participants, most of them being parents of OTC patients. As the subject is a rare disease, this number is not small, and the results can be considered significant and a basis for more extensive further studies. 

The manuscript is well written in general, following the most important issues on this important subject for a rare disease. There is no need for corrections regarding the use of the English language. There are only some minor improvements that may be necessary: to have larger figures, not to include the results in discussions again, and to move some ideas from conclusions to discussions (that with citation).

Author Response

Thank you for the review and the comments, we have enlarged the figures and removed all citations from the conclusion section. The passages in the discussion in which results were mentioned again have been revised.

Round 2

Reviewer 1 Report

Thanks for having gone through the comments and for having modified the manuscript accordingly.